# LAVA: Long-horizon Visual Action based Food Acquisition

Amisha Bhaskar, Rui Liu, Guangyao Shi, Pratap Tokekar

*Abstract*— Robotic Assisted Feeding (RAF) addresses the fundamental need for individuals with mobility impairments to regain autonomy in feeding themselves. The goal of RAF is to use a robot arm to acquire and transfer food to individuals from the table. Existing RAF methods primarily focus on solid foods, leaving a gap in manipulation strategies for semi-solid and deformable foods. This study introduces Long-horizon Visual Action (LAVA) based food acquisition of liquid, semisolid, and deformable foods. Long-horizon refers to the goal of "clearing the bowl" by sequentially acquiring the food from the bowl. LAVA employs a hierarchical policy for long-horizon food acquisition tasks. The framework uses high-level policy to determine primitives based on food types. At the mid-level, LAVA finds parameters for primitives using vision. To carry out sequential plans in the real world, LAVA delegates action execution which is driven by Low-level policy that uses parameters received from mid-level policy and behavior cloning ensuring precise trajectory execution. We validate our approach on complex real-world acquisition trials involving granular, liquid, semisolid, and deformable food types along with fruit chunks and soup acquisition. Across 46 bowls, LAVA acquires much more efficiently than baselines with a success rate of $89 \pm 4\%$, and generalizes across realistic plate variations such as different positions, varieties, and amount of food in the bowl. Code, datasets, videos, and supplementary materials can be found on our website.

## I. INTRODUCTION

For individuals limited mobility or disabilities, self-feeding can be a daunting task, underscoring the need for Robotic Assisted Feeding (RAF) [1] systems, to enhance independence and quality of life as well as reducing caregiver burden. Dealing with various foods—from granular cereals to semi-solid food such as yogurt and deformable food items such as tofu, without breakage or deformation presents significant challenges for RAF [2], [3]. Traditional RAF methods have relied on pre-set strategies for specific tasks like skewering [4]–[7], bite transfer [4], [8], [9], and scooping [2], [10], which falls short in complex feeding scenarios akin to human feeding actions. This gap highlights the need for replicating nuanced, human-like feeding strategies. This gap in technology prompts the exploration of hierarchical frameworks that break down intricate feeding actions into simpler steps [7], [11]–[13], addressing the challenge of complex food handling. Yet, deploying these frameworks to manage the diverse and changeable nature of food in real-world settings remains a formidable challenge. We aim to leverage hierarchical planning, vision-based control, and flexible adaptation to various food types, addressing the limitations of current RAF approaches.

All authors are from the University of Maryland, College Park, MD 20742 USA. {amishab, ruiliu, gyshi, tokekar}@umd.edu

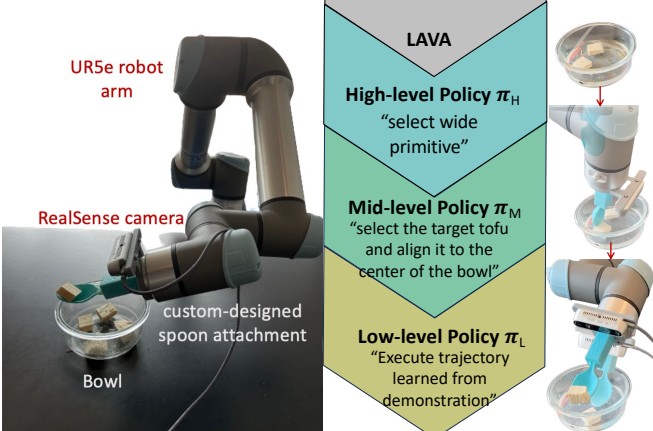

Fig. 1: System setup for LAVA with an illustrative description of the proposed framework with snapshots of task execution.

## II. PROBLEM STATEMENT

This study tackles the challenge of sequential bite acquisition to maximize the success rate and efficiency of long-horizon food acquisition for efficient bowl clearance. The focus is on a variety of food types, from granular items such as cereals to semi-solid foods such as yogurt, and deformable substances such as tofu, all within a static bowl and assumed to be scoopable with a spoon. We assume access to bowl image observations $o \in \mathbf{R}_+^{W \times H \times C} = \mathcal{O}$ of unknown bowl states $S$. Here, $W$, $H$, and $C$ denote the image dimensions. The image is sourced from a camera attached to the wrist of the robotic arm. We have access to expert demonstration data for robot proprioceptive information (joint positions). Our goal is to learn a policy $\pi(\phi_t|o_t)$ that takes RGB images as input $(o_t)$ and returns output as joint angles $\theta_t$ of the arm for efficient long-horizon food acquisition.

## III. PROPOSED APPROACH

We formalize the long-horizon food acquisition setting as a hierarchical policy $\pi$. To do so we decouple $\pi$ into separate high, mid, and low-level sub-policies. We assume access to $K$ discrete manipulation primitives $P_H^k$, $k \in 1, ..., K$, and learn a high-level policy $\pi_H$ which selects amongst these primitives based on visual input $o_t$. The mid-level policy $\pi_M$ further refines this selection, parameterizing the low-level policy $\pi_L$ based on both the chosen primitive and additional visual inputs. This low-level policy then executes a sequence of actions $\theta_t^k$, aimed at achieving precise food acquisition. The formulation of this hierarchical arrangement is as follows:

- **High-level policy:** $\pi_H(P_H^k|o_t)$ focuses on selecting the manipulation primitive for the current observation.

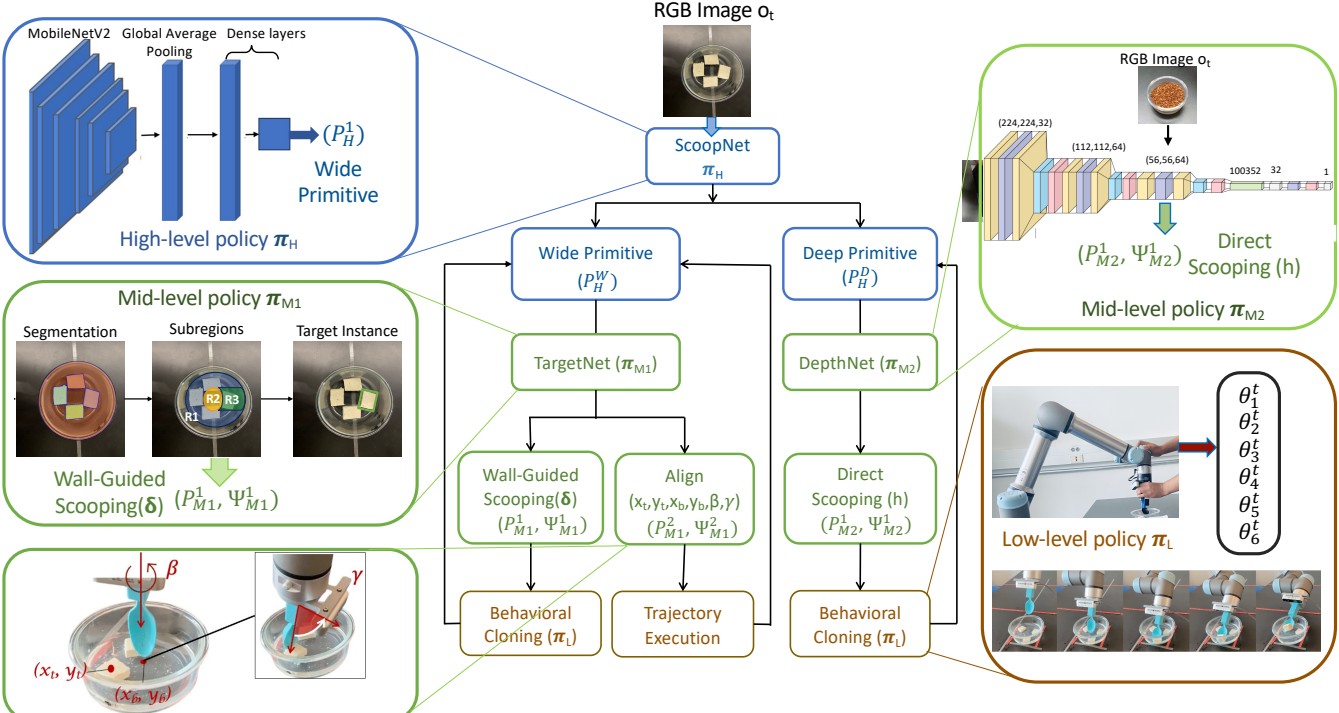

Fig. 2: System Architecture of LAVA, employs a high level policy(blue) $\pi_H$ to select amongst discrete high level primitives $P_H^k$, which further gets refined by mid-level policy (green) $\pi_M$ to select amongst mid-level primitives $P_M^k$, low-level vision parametrized policy $\pi_L$ (brown) executes trajectory learned from Behavioral cloning for long-horizon food acquisition.

- **Mid-level policy:** $\pi_M(P_M^k, \psi_M^k|o_t, P_H^k)$ refines this choice by parameterizing actions to the specific food item's characteristics.
- **Low-level policy:** $\pi_L(\theta_t^k|P_M^k, \psi_M^K)$ executes the action sequence, using parameters from the mid-level policy.

We consider low-level actions $\theta_t$, parameterized by the position of the tip of a spoon $(x, y)$ and spoon roll and pitch $(\gamma, \beta)$ in the wrist frame of reference. As shown in Figure 2 detailing the LAVA setup, for more detailed description of each module, refer to our paper [14].

### A. High-level Policy

At the highest level of our hierarchical model, the high-level policy $\pi_H(P_H^k|o_t)$ uses visual cues to select the most suitable scooping primitive—Wide Primitive ($P_H^W$) for non-cohesive, deformable foods such as tofu, and Deep Primitive ($P_H^D$) for cohesive foods such as cereals. The Wide Primitive leverages the bowl's wall for support, creating a mass easy to scoop without causing food breakage, while the Deep Primitive enables direct scooping with precise control over spoon trajectory for minimal disturbance.

ScoopNet ($\pi_H$): ScoopNet, built on the MobileNetV2 architecture [15] as the base, distinguishes between these primitives. Trained on a dataset of 5316 images for accurate primitive selection, employing a Global Average Pooling layer and dense layers for refined classification. We use Adam optimizer and binary cross-entropy loss for the Optimizations, producing softmax probabilities for selecting scooping strategies for specific task adaptation.

For more detailed information, refer to our paper [14].

### B. Mid-level Policy

The Mid-level Policy $\pi_M(P_M^k, \psi_M^K|o_t, P_H^k)$ refines and parameterizes the chosen primitive, crucial for translating high-level strategy decisions into low-level action execution.

*1) TargetNet ($\pi_{M1}$) for Wide Primitive:* TargetNet employs Mask R-CNN to identify and segment target items such as tofu for scooping. This model segments food items, enabling the selection of appropriate mid-level primitives: wall-guided scooping and center align using annotations for precise segmentation and transfer learning for accuracy. TargetNet divides the bowl into sub-regions (R1 for rightmost and closest to the wall, R2 for center and R3 otherwise) to guide scooping decisions, whether leveraging wall support or aligning for easier access [14].

**Wall-guided Scooping and Align**: This method varies scooping based on food's position—Wall-guided Scooping for foods in subregions R1 and R2 and Align for food positioned in R3 subregion. The alignment step calculates the spoon's orientation and the distance to move food towards the bowl's center, optimizing scooping paths.

*2) DepthNet ($\pi_{M2}$) for Deep Primitive:* DepthNet, with its Sequential model, determines food depth in the bowl, aiding in selecting the depth for deep scooping of cohesive foods. It's trained on diverse cereal images to precisely estimate food volume, adjusting the scooping depth accordingly for effective clearance [14].

**Direct scooping** ($P_{M2}^1, \psi_{M2}^1$) Incorporates real-time feedback to adjust scooping strategies based on DepthNet's depth

information, using behavior cloning to refine the spoon's path, ensuring efficient scooping across different food depths.

## C. Low-level policy

At the foundation of our model, we use behavioral cloning ($\pi_L$), coupled with kinesthetic teaching [16], to fine-tune the robot's scooping actions across varied food textures, directly informed by expert demonstrations. This method involves learning distinct scooping trajectories for different foods. The goal is to minimize deviations from these optimal paths using a cost function $J(\tau)$, with the Weiszfeld algorithm [17], [18] applied for optimization. This algorithm iteratively adjusts the estimated trajectory $\hat{x}$, improving scooping precision by reducing the sum of distances from demonstrated trajectories until minimal changes are achieved. For a deeper dive into the specifics of our behavioral cloning approach and its application within **LAVA**, refer to our discussion in [14].

## IV. QUANTITATIVE RESULTS

Our experiments detailed in [14] involve a comprehensive setup (see Figure 1) including a UR5e robot arm with custom spoon attachment, and a RealSense camera, testing on a variety of food types from cereals to tofu in soup. Utilizing two baselines, LAVA-low and Fixed Trajectory Scooping, for comparative analysis, we explore a range of food configurations to assess our hierarchical framework's effectiveness in adaptive food acquisition. Our key findings:

*1) Network Performance:* ScoopNet achieved 100% accuracy in choosing correct high-level primitives, TargetNet accurately predicted bite targets at 87.9% , and DepthNet successfully determined correct spoon depths for bite sizes at 85.7%, demonstrating the **LAVA** networks' effectiveness in robotic-assisted feeding.

*2) Baselines Comparison:* **LAVA** outperformed both baseline models, LAVA-low and FTS, in efficiency, scoop size, and minimizing spillage and breakage as visible in Figure 4 and Figure 3.It adeptly managed liquids, significantly minimized breakage with deformable foods such as tofu through strategic scooping, and ensured minimal spillage with solid foods using align-then-scoop strategy.

*3) Zero-shot Generalization:* **LAVA** effectively handled diverse foods, including soup with tofu and apple chunks, showcasing adaptability in Figures 3 and 5. Its ability to adjust in real-time for both solid and liquid scooping underlines **LAVA**'s robustness across food types.

For a comprehensive overview of **LAVA**'s methodologies, outcomes and adaptability, see [14].

## V. CONCLUSION, LIMITATION AND FUTURE WORK

In this study, we introduced a hierarchical policy framework, **LAVA**, that improves robotic food acquisition from liquids to deformable solids. Utilizing **LAVA**'s networks, it addresses the variability in food types, achieving higher efficiency and accuracy with less spillage and breakage than baselines. Despite its success, challenges remain with thin or irregularly shaped foods. Future work aims to expand the action space and explore new data acquisition methods, potentially using online videos for complex food interactions.

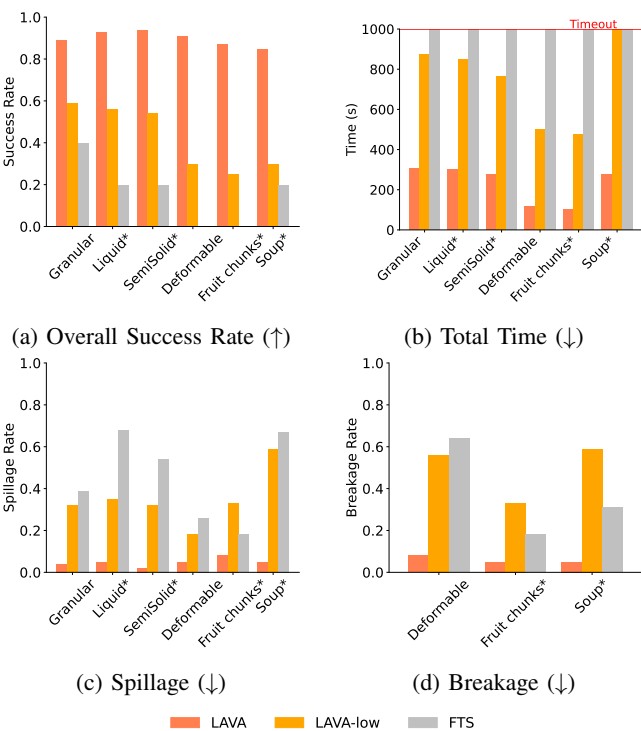

(a) Overall Success Rate (↑)   (b) Total Time (↓)

(c) Spillage (↓)   (d) Breakage (↓)

LAVA   LAVA-low   FTS

Fig. 3: Breakdown of experimental performance comparison between **LAVA**, LAVA-low, and Fixed Trajectory Scooping(FTS). ∗ represents zero-shot experiments.

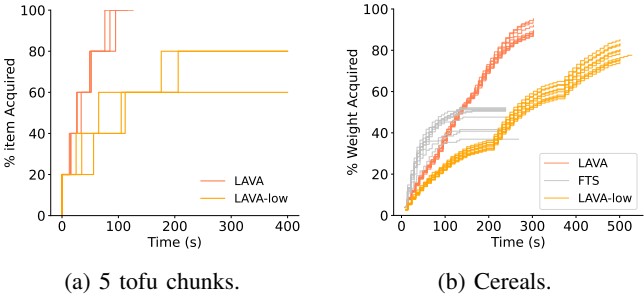

(a) 5 tofu chunks.   (b) Cereals.

Fig. 4: Individual trials comparison between **LAVA** and baselines: (a) different tofu configurations, (b) cereals

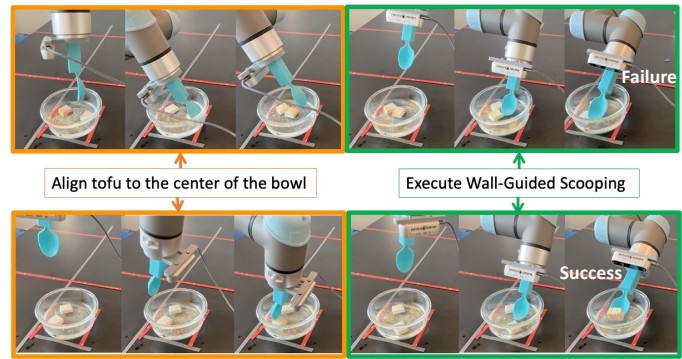

Fig. 5: Zero-shot acquisition with tofu in soup: Top images depict spoon alignment of tofu to the bowl's center, which drifts due to soup's fluidity. Bottom images show realignment and successful scooping.

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
