# OpenReview forum: "LAVA: Long-horizon Visual Action based Food Acquisition"
_IEEE.org/2024/ICRA/Workshop/CookingRobot — CookingRobot2024 Poster_

### Official Review · Reviewer_6ZHq · 2024-04-11
**The review for LAVA: Long-horizon Visual Action based Food Acquisition**

**Rating:** 10
**Confidence:** 5

**Review:**

*Major Contribution of the Paper:

This paper presents a hierarchical policy framework designed for robotic-assisted feeding to handle various food types including liquids, semisolids, and deformable solids. Their framework, LAVA utilizes a combination of high-level, mid-level, and low-level policies to address the complexity of food acquisition tasks, achieving efficient and adaptable performance across different bowl configurations and food types.


*Major comments:

The paper is addressing a critical challenge in handling diverse food types with robotic arms, which is necessary for cooking robots. I am looking forward the further discussion in the workshop.

It is a realistic and efficient idea to branch out primitives for each type of food and tailor actions accordingly. However, I have concerns that when dealing with other completely different types of food, such as flat and thin items or heavy and large ones, further branching will be necessary, potentially requiring a massive framework. I believe that further generalizing the current framework would lead to even greater improvements.

Using vision to discriminate between food types and predict the depth of granular items is valuable, however, I questioned if there may be situations where visual cues alone are insufficient for accurate judgments. Additionally, I am concerned about the framework's ability to adapt when backgrounds or containers change.

*Video:

Nice scooping demonstration. The motion is smooth and accurate.
And the video helps understanding the transition of the hierarchical level well.

*Paper

The paper covers an overall good summary of the method. The results with success rate using various ingredients are well-presented.

---

### Official Review · Reviewer_b4PU · 2024-04-16
**The review for ”LAVA: Long-horizon Visual Action based Food Acquisition”**

**Rating:** 9
**Confidence:** 4

**Review:**

This paper addresses the task of Robotic Assisted Feeding (RAF) and proposes a hierarchical framework that can accommodate liquid, semi-solid, and deformable foods.
Manipulation of diverse and deformable foods is an important issue in the field of cooking robots and surrounding food-related robotics. Therefore, the proposals and experimental results of this paper will bring the discussion at the workshop to a higher level.

Major comment:
* Further information on the limitations of this study and its solutions would be useful. As mentioned in the paper, there are still problems with thin or irregularly shaped foods, and it would be important knowledge on the actual cases of failure, their causes, and possible solutions. In particular, it is important whether the number of primitives needs to be increased and what other primitives can be considered.

Video:
*  The video is very clear about the proposed hierarchical policy framework and the experiment. It is very good to see the robot scooping up the food after aligning it.